# A Multi-Center Disclusion Time Reduction (DTR) Randomized Controlled Occlusal Adjustment Study Using Occlusal Force and Timing Sensors Synchronized with Muscle Physiology Sensors

**DOI:** 10.3390/s21237804

**Published:** 2021-11-24

**Authors:** Prafulla Thumati, Roshan P Thumati, Shwetha Poovani, Atul P Sattur, Srividya Srinivas, Robert B Kerstein, John Radke

**Affiliations:** 1Department of Prosthodontics, Raja Rajeswari Dental College and Hospital, Bengaluru 560074, India; thumatiprafulla@gmail.com (P.T.); drshwetapoovani@yahoo.com (S.P.); 2Postgraduate Orthodontics and Dentofacial Orthopedics Departments, Oxford Dental College, Bengaluru 560068, India; ro.sh.jaglur@gmail.com; 3Department of Oral Medicine & Radiology, SDM College of Dental Sciences and Hospital, Sattur, Dharwad 580009, India; atulsattur@gmail.com; 4Department of Prosthodontics, AECS Maruti Dental College and Research Centre, Bengaluru 560076, India; drsrividya5@gmail.com; 5Department of Restorative Dentistry, Tufts University School of Dentistry, Boston, MA 02111, USA; 6Chairman BioResearch Associates, Inc., Milwaukee, WI 53223, USA; jradke@bioresearchinc.com

**Keywords:** High Definition (HD) Novus sensor, bipolar EMG leads, Disclusion Time Reduction (DTR), Immediate Complete Anterior Guidance Development Coronoplasty (ICAGD), beck depression inventory II, pain frequency scale, pain intensity scale, functional restriction scale

## Abstract

**Objective**—To perform a Randomized Controlled Trial (RCT) Disclusion Time Reduction (DTR) study at five Dental Colleges, using intraoral sensors and muscular electrodes. **Methods and Materials**—One hundred students were randomly assigned to a treatment group to receive the ICAGD coronoplasty, or a control group that received tooth polishing. All subjects answered symptom questionnaires: Beck Depression Inventory-II, Functional Restrictions, and Chronic Pain Symptom and Frequency. Subjects self-reported after ICAGD or placebo at 1 week, 1 month, 3 months, and 6 months. The Student’s t-Test analyzed the measured data. The Mann–Whitney U Test analyzed the subjective data (Alpha = 0.05). **Results**—The Disclusion Times, BDI-II scores, and Symptom Scales were similar between groups prior to treatment (*p* > 0.05). At 1 week, all three measures reduced in the treatment group, continuing to decline over 6 months (*p* < 0.05), but not for the controls (*p* > 0.05). Symptom Frequency, Functional Restrictions, and Pain Frequencies were higher in the treated group (*p* < 0.05), but declined after ICAGD compared to the control group (*p* < 0.05). **Conclusions**—ICAGD reduced Pain, Functional Restrictions, Symptom Frequency, and Emotional Depression within 1 week, which continued for 6 months. The tooth polishing did not initiate a placebo response.

## 1. Background

The T-Scan Novus High Definition (HD) sensor (TekScan Inc. S. Boston, MA, USA) is an ultra-thin (0.004″ or 0.1 mm thick), flexible printed circuit that records patient relative occlusal forces and contact timing sequences in 0.003 s/frame when crushed between teeth [1]. The large sensor (#2002) has 1370 active pressure sensing locations, while the small sensor (#2502) has 1122 (Figure 1). The HD sensor consists of two layers of Mylar encasing a printed grid of sensing locations (referred to as “sensels”), which are arranged in rows and columns in the form of a dental arch (Figure 2).

The HD sensors lock into the Novus recording handpiece that connects to a computer via a USB interface (Figure 1). The inner handpiece electronics mate with the sensor tab’s conductive components (Figure 2; Tab Length A) to relay the interarch dental occlusal force and timing data to companion software using a multi-color two- and three-dimensional dynamic video display (Figure 3). The HD sensor can be clenched into or chewed over by a patient, to record changing tooth-to-tooth relative occlusal contact force interactions in real-time, as they fluctuate during functional mandibular movements.

As opposing teeth make occlusal contact, the upper and lower sensor surfaces are compressed, which changes the resistance in each loaded sensel. Higher applied contact force produces larger resistance changes, and lower occlusal contact force produces lesser resistance changes. These resistance changes are measured in *Digital Output Voltages (DO)*. Higher occlusal forces give off higher digital output, while lower applied forces give off lower digital output [1].

The recording handpiece electronics read the output voltages of each row of sensels in a process called *Scanning*. The sensor’s output signal is conditioned and converted to an 8-bit digital value, so that each measured resistance change is proportional to the force applied to each sensel, with a range between 0 and 255 raw counts. The total force measured is the aggregate of all raw force outputs from each sensel, and it is reported as the *Raw Sum* force. The raw count from each individual 1.61 mm^2^ sensel can be conceptualized as a force [2]. A map in the companion software (T-Scan software version 10, Tekscan Inc., S. Boston, MA, USA) receives a stream of changing sensel digital output values and organizes them for display in the same orientation that the sensels have on the sensor grid [2]. The T-Scan software’s multi-colored two- and three-dimensional graphical display presents the variable digital outputs for clinician occlusal analysis (Figure 3), which has been shown in many published studies performed with Electromyography to aid clinicians in diagnosing and treating many occlusal abnormalities and their non-physiologic muscular responses [3,4,5,6,7,8,9,10,11,12,13,14,15,16,17,18,19,20,21].

A unique application of the T-Scan system is that it can be synchronized to record simultaneously with a head and face muscle Electromyography system (BioEMG III, Bioresearch Assoc., Milwaukee, WI, USA) (Figure 4 and Figure 5) [5]. The synchronization records transitory occlusal contacts that are time-correlated to changes in the electrical potential of up to eight masticatory muscles. This synchronization has been used in occlusal adjustment and muscle physiology studies to assess occlusal function and its associated muscle contraction levels [3,4,5,6,7,8,9,10,11,12,13,14,15,16,17,18,19,20,21]. However, to date, the T-Scan/BioEMG synchronization has not been employed in a Randomized Controlled (RCT) multi-center trial occlusal adjustment study.

A multicenter randomized controlled trial (RCT) is a clinical trial conducted at more than one medical center or clinic [22,23,24]. The benefits of multicenter trials include a larger number of participants from different geographic locations, the possibility of including a wider range of population groups, and the ability to compare results between centers, all of which increase the generalizability of the study [22,23,24].

Chronic Occluso-muscular TMD is characterized by pain and noises in the Temporomandibular Joint, frequent temporal headaches, neck pain, ear pain, chewing fatigue, clenching and grinding habits, morning facial muscle pain and stiffness, and restricted mouth opening [7,25,26]. The role of occlusal adjustment in the reduction of chronic orofacial pain symptoms historically has been “very controversial.” A few controlled studies (not RCTs) have shown improvement of symptoms in the treated groups [7,25,26], while other authors have found little or no correlation between occlusal contacts and orofacial pain symptoms [27,28,29].

A known correlation exists between posterior tooth contact time durations in mandibular excursive movements and masticatory muscle activity levels [4,30,31,32]. The longer in time the posterior teeth remained in contact during mandibular excursions (known as the *Disclusion Time*) [4,32], the longer were the tooth flexure durations, the pulpal nerve flexures, and the periodontal ligament mechanoreceptor nerve compressions of the opposing contacting teeth, which, through the posterior tooth neural feedback loop [33], adds excess muscle contractions to the baseline muscle contractions needed to perform mandibular movements. This neuro-physiologic mechanism has been shown in studies to cause muscle hyperactivity and increased muscle contraction durations [6,7], which delays the involved muscles from reaching low-level contractions during the excursive movements, accumulating lactic acid that leads to ischemic pain and dysfunctional masticatory muscles.

A recent RCT DTR occlusal adjustment study reported that chronic painful muscular TMD symptoms, Functional Restrictions, and levels of Emotional Depression from living with chronic symptoms were all improved in the treatment group within one week after receiving ICAGD, and continued to improve during the remainder of the study [13]. The mock ICAGD placebo did not initiate a placebo effect in the control subjects, because tooth polishing made no physical changes to the control group’s Disclusion Times [13]. Importantly, ICAGD has known high-tolerance numerical outcome measures [7,34] that make it a highly precise occlusal adjustment procedure compared to the subjective and unmeasured procedure known as *Occlusal Equilibration* [34].

To date, there are no published multi-center Randomized Controlled ICAGD/DTR occlusal adjustment studies. As such, the objective of this multi-center research was to perform an RCT Disclusion Time Reduction study employing HD Novus sensors and BioEMG III adhesive bipolar electrodes at five different treatment centers. This study design could validate that different clinicians performing the same measured ICAGD coronoplasty in differing locations could potentially obtain statistically similar and verified chronic muscular TMD symptom improvements, which have been observed in many prior single-center Disclusion Time Reduction studies [3,4,5,6,7,8,9,10,11,12,13,14,15,16,17,18]. 

## 2. Materials and Methods

Five calibrated Disclusion Time Reduction clinicians located at four treatment centers within India were recruited to participate. Two treating clinicians were located at the Oro-Facial Pain Clinic of Raja Rajeswari Dental College and Hospital in Bengaluru, one at the Orthodontic and Dentofacial Orthopedics Department of Oxford Dental College in Bengaluru, one at the Department of Oral Medicine and Radiology at SDM College of Dental Sciences and Hospital in Sattur Dharwad, and one at the Department of Prosthodontics at AECS Maruti Dental College and Research Centre in Bengaluru.

The same inclusion/exclusion criteria and randomization procedure were employed at all five treatment centers to develop, per center, a group of 10 treatment subjects and 10 control subjects. This yielded a combined *n* = 50 Treatment subjects and *n* = 50 Control subjects. All potential subjects answered a questionnaire to qualify their possible study participation, comprised of questions regarding their TMD history, symptoms, prior treatments, and daily symptom frequency and intensity. Each potential subject also completed a Beck Depression Inventory-II (BDI-II) [35,36,37,38] to classify their emotional depression level related to living with chronic painful symptoms.

### 2.1. The Inclusion Criteria

A history of chronic muscular myofascial pain dysfunction symptoms;A fully dentulous state of at least 28 teeth;Near normal occlusal relations with opposing molar and premolar teeth in contact in right and left excursions;Angles Class I and Class III subjects with anterior guiding teeth that were either in contact, or near to contact.

### 2.2. The Exclusion Criteria

Severe Class II and Class III malocclusions;Large anterior open occlusions absent of anterior guidance contacts;A previous history of trauma to the TMJ region;Patients who had undergone prior treatment for Temporomandibular Disorder;Patients who had undergone prior occlusal adjustment treatment.

One hundred total subjects were selected (20 from each center), who all demonstrated medium to high Beck Depression-II scores, all with Angle’s Class 1 (or Class III with anterior guiding teeth), skeletal relations, and 28 teeth, and who were experiencing mainly muscular and only minor Temporomandibular Joint symptoms. The subjects were assigned to the treatment or control groups non-alphabetically by pulling their names out of a hat. The first name chosen was placed into the treatment group, and the second name was placed into the control group. Each subsequent name was then alternated into the treatment group followed by the control group until all 100 subjects were divided into the two groups. 

The treated group mean age (21 +/− 1.87 years) was significantly older (*p* < 0.05) than the control group mean (19 +/− 1.21 years). 12 subjects in the treatment group and 17 subjects in the control group had prior Orthodontic treatment. All subjects filled out an Informed Consent that stated the participants would not be told if they were to receive the treatment or the placebo (mock ICAGD), but that both groups would have their enamel adjusted and polished. The protocol was reviewed and approved by the Institutional Ethical Committee at Raja Rajeswari Dental College and Hospital, Bengaluru, India, and given Clearance Number RRDCH/300/20-21.

### 2.3. Pretreatment or Preplacebo Clinical Photogrpahs

Clinical photographs of each subject’s MIP and their right and left translated occlusal relations were obtained prior to any ICAGD or mock ICAGD being rendered (Figure 6, Figure 7 and Figure 8). 

### 2.4. Pretreatment or Preplacebo T-Scan 10/BioEMG III Data Aquisition

The synchronized T-Scan/EMG was then used to record pretreatment or pre-placebo multi-bite closure-into-MIP data (Figure 9), and the right, left, and protrusive excursions of all subjects. Excursive recordings were accomplished by subjects closing into Maximum Intercuspal Position (MIP) firmly, holding their teeth together for 1–3 s, and commencing a right/left/protrusive excursion until only anterior teeth were in contact (Figure 10 and Figure 11; protrusion not shown for brevity). This specific recording method ensures that high quality Disclusion Time and EMG data is obtained from the subjects [1].

The Pre-treatment Disclusion Times values and excursive electromyography levels were recorded for comparison to the post-ICAGD coronoplasty values and EMG levels in the Treated group, and for post-placebo polishing comparison in the Control subjects. All the pre- to post-ICAGD or Placebo Disclusion Time/EMG data was compared at Day 1 before to after ICAGD or placebo, then at Week 1, Month 1, Month 3, and Month 6. 

### 2.5. ICAGD Occlusal Adjustment Procedure Performed on the Treated Group

The subjects’ teeth were air dried. The patients were then asked to close into their Maximum Intercuspal Position (MIP) with articulating paper (Arti-Fol^®^ Red, 8μ, Bausch GMBH, Hainspitz, Germany) interposed between their teeth, and then to commence a right mandibular excursion out to their right canine edges, then slide back into MIP, and then make a left mandibular excursion out to their left canine edges, and then back into MIP. 

The pre-treatment T-Scan/BioEMG recordings guided the areas to adjust occusally, where prolonged excursive frictional contacts were detected in the resultant articulating paper markings representative of lengthy Disclusion Time (Figure 12; inclined plane linear contact patterns). The molar and pre-molar excursive contacts were eliminated on all involved surfaces using pear shaped finishing burs (Mani Dia-Burs, Japan ISO no-237/021) bilaterally, leaving the central fossa, cusp tip, and the marginal ridge contact points intact.

### 2.6. ICAGD Was Considered Completed When

All Class I, II, and III lateral excursive interferences had been visually removed;Disclusion of all posterior teeth in the right and left excursions was visible with the subject experiencing easier lateral movements than pre-ICAGD (Figure 13);All linear contacts had been removed;The remaining pattern of habitual closure contacts were located solely on cusp tips, fossae, and marginal ridges (Figure 14).

**Figure 13 sensors-21-07804-f013:**
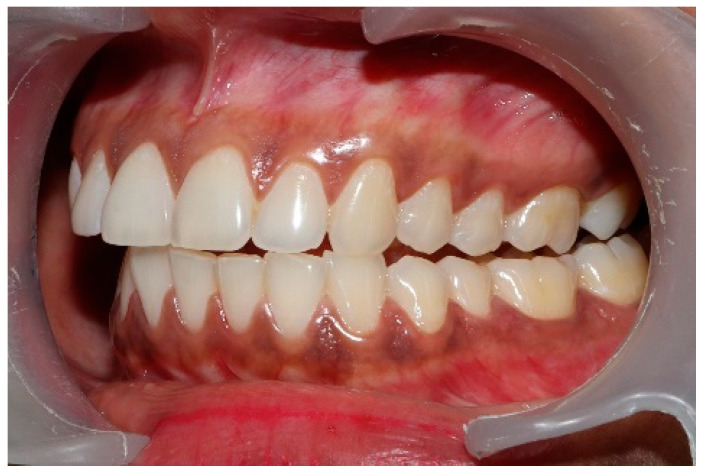
Visible left posterior disclusion with canine control over the lateral movement post-ICAGD. By reshaping the posterior teeth with the ICAGD protocol, disclusion was achieved despite the worn maxillary left canine.

**Figure 14 sensors-21-07804-f014:**
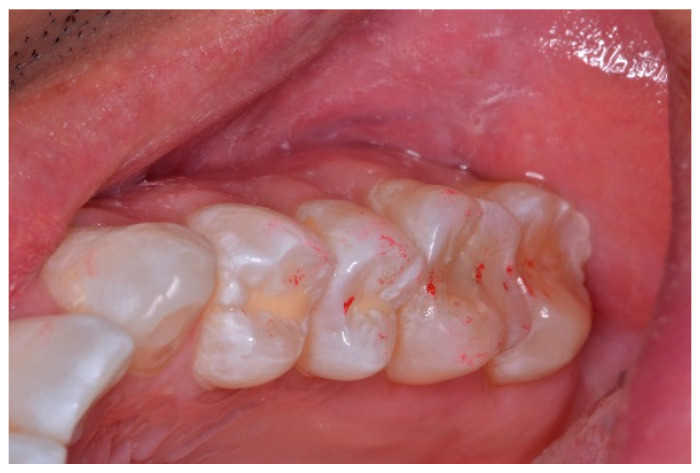
Upper left quadrant of articulating paper markings post-ICAGD, representative of short Disclusion Time. There is no longer marked interocclusal surface engagement and friction present when the patient excurses right, left, or forwards.

A new set of T-Scan 10/BioEMG III recordings of each excursion were then performed to verify that the Disclusion Times had been reduced to <0.5 s. Any remaining prolonged excursive contacts were adjusted until each excursion’s Disclusion Time was less than 0.5 s (Figure 15 and Figure 16). 

The treated ICAGD group had their Disclusion Time T-Scan/BioEMG data recorded once per day at Week 1, Month 1, Month 3 and Month 6 recall intervals, at which time the treated subjects self-reassessed their symptoms with new questionnaires. This completed the treatment phase.

### 2.7. Procedures Performed for the Placebo Group—Mock ICAGD 

Pre-mock and post-mock ICAGD and the Disclusion Time T-Scan/BioEMG recordings were made in the same fashion as was done with the treated group. The patients also performed excursions and unguided closures through the articulating paper in the same fashion as did the treated group. The resultant excursive contact patterns were polished with a rubber cup and polishing paste 3–5 times per arch, to mimic the true adjustments performed on the treated group. The polishing was completed when the control subjects “felt” their polished contacts were balanced and comfortable. Importantly, the control subjects were blinded to the handpiece, which was always covered by cloth. 

The placebo ICAGD group had their Disclusion Time T-Scan/BioEMG data recorded once per day at Week 1, Month 1, Month 3, and Month 6 recall intervals, at which time the mock adjustment subjects self-reassessed their symptoms with new questionnaires. 

No complimentary therapies and no pain medications were applied to either the control group or the treatment group. The only treatments applied were the ICAGD coronoplasty to the treated group, and tooth polishing as the “placebo ICAGD” to the control group, and no other uncontrolled variables were identified. 

All of the T-Scan/EMG data from the five test sites was collated and verified by the Principal Investigator (PT) to ensure that the Quadrant Division Line in the two-dimensional T-Scan Force View, and the Force vs. Time graph C and D lines per subject per date of recording, were all properly placed (as previously described in a published appendix (3) to report accurate Disclusion Time values for statistical analysis. These Disclusion Times pre- and post-ICAGD and placebo ICAGD, and the excursive EMG muscle activity level data pre- and post-ICAGD and placebo ICAGD were subjected to the Student’s paired t-Test (Alpha = 0.05). 

### 2.8. Method of Symptom Assessment

Patient self-assessment with repeated ordinal number symptom frequency and intensity questionnaires were used pre-treatment, and at all dates of treatment or placebo. This subjective questionnaire data underwent the non-parametric Mann–Whitney U test for inter-group comparisons and the Wilcoxon signed-rank test for intra-group comparisons (Alpha = 0.05). Although subjects received treatment/placebo on different days, the analyses were first carried out at 1 week to assess the immediate response to treatment/placebo, and then repeated at 1 month, 3 months, and 6 months from the starting date of each patient to assess whether the response was sustained. Because all treatment subjects and control subjects were assessed at the same relative timepoints, no time adjustment was indicated.

## 3. Results

### 3.1. Disclusion Times 

Prior to any treatment or mock ICAGD, there was no significant difference between the treatment and control Disclusion Times, but at all follow-up dates, the treatment Disclusion Times were significantly less than the controls (*p* < 0.05) (Table 1).

### 3.2. Mean BDI-II Scores

There was no significant difference between the mean BDI-II scores of the control group 19.3 (+/−7.26) and the Treated Group 20.1 (+/−4.98) prior to any treatment or placebo (Mann–Whitney U test *p* = 0.5222). One week after ICAGD, the mean BDI-II score was reduced to 4.1 (+/−2.03), while the control group mean BDI-II score had increased somewhat to 23.0 (+/−6.62) (Mann–Whitney U test *p* < 0.00001). The mean BDI-II scores for the treated group continued to decline over time, and by 6 months’ post-ICAGD reached 0.78 +/− 0.98. At the same time-point, the control group mean BDI-II score remained high at 24.9 +/− 6.02 (Mann–Whitney U test *p* < 0.00001) (Table 2).

### 3.3. Mean Symptom Pain Scale Scores

The mean Symptom Pain Scale score for the control group, 4.9 (+/−4.28), was lower prior to placebo treatment than the treated group’s mean score of 6.6 (+/−5.96), but not significantly (*p* = 0.238). The mean Symptom Pain Scores of the treated group were reduced at one month post-ICAGD to 0.06 (+/−0.24) and then stabilized up to the 6 month time point. In contrast, the control group’s mean Symptom Pain Scale score increased over the same time-period to 7.6 +/− 4.99 (Table 3). 

### 3.4. Mean Frequency of Symptoms Scores

The mean Frequency of Symptoms score for the control group, 3.7 (+/−3.78), was significantly lower prior to placebo than the treatment group’s mean score of 8.8 (+/−4.1) (*p* < 0.05). At Week 1, there was no significant difference (*p* = 0.1868), but at Month 1, the treatment group mean score had reduced significantly to 0.02 (+/−0.14) (*p* < 0.00001) (Table 4).

### 3.5. Mean Functional Restrictions Scores

The mean Functional Restrictions score for the control group, 4.5 (+/−2.95), was significantly lower prior to treatment than the treatment group’s mean score of 7.8 (+/−2.73) (*p* < 0.05). At Week 1 post-ICAGD, the treatment group’s score had reduced significantly to 0.7 (+/−1.41) (*p* < 0.00001), while the control group’s score remained high at 4.9 (+/−3.05). This relationship remained throughout the 6-month follow-up (Table 5).

### 3.6. Mean Frequency of Painful Symptoms Scores

The mean Frequency of Painful Symptoms score for the control group, 5.80 (+/−4.65), was significantly lower (*p* < 0.05) prior to placebo than the treatment group’s mean score of 19.5 (+/−5.51). By Week 1 post-ICAGD, the treated group’s Functional Restriction score, 4.04 (+/−1.67), was significantly lower than the control group’s score 6.1 (+/−4.68) (*p* = 0.0043) (Table 6). 

The overall Results strongly indicate the placebo treatment did not initiate a placebo effect in the control subjects. The trends of the treated subjects’ improvements and the control’s non-improvements greatly differed (Figure 17).

## 4. Discussion

### 4.1. Corroboration of Prior Findings

The Results of this multi-center ICAGD RCT study do corroborate the findings of the prior single-center RCT ICAGD/DTR study, and the previous non-RCT ICAGD/DTR studies, where marked and rapid symptom improvements were reported following ICAGD [3,4,5,6,7,8,9,10,11,12,13,14,15,16,17,18]. This presented study tracked symptom intensities and frequencies, and their emotional impact on subjects. Many painful symptoms improved within one week after ICAGD was rendered, which led to significant reductions in the treated subjects’ levels of depression, which continued to improve over the 6-month period of observation. 

Symptom resolution for most treated subjects occurred within the first month after their *long* Disclusion Times were reduced to *short* Disclusion Times. Symptoms rapidly lessened because the pre-treatment elevated levels of muscle activity that cause lactic acid buildup, and its resultant ischemic pain and functional limitations of the masticatory muscles, was markedly lessened after ICAGD removed the occlusal surface friction from the subject’s mandibular excursions. These physiologic changes are neurologically-mediated within the Central Nervous System [33], because the time-duration and volume of posterior teeth pulpal nerve fiber flexures and PDL nerve fiber compressions are both drastically reduced by ICAGD, making the treated subjects function muscularly at or near resting state contraction levels [4,5,6]. These low-level excursive movement contractions stop lactic acid from building up, which limits the resultant ischemia, allowing for the lactic acid to be metabolized away, instead of it continuing to accumulate. Without ischemia present, the masseter and temporalis muscle fibers can re-oxygenate, which lessens muscular pain, reduces the frequency of symptom appearances, and resolves functional limitations, whereby chewing function dramatically improves [3]. This ICAGD-induced absence of muscular ischemia is quickly followed by lasting symptom resolution [10,17,39].

This study’s results also strongly correlate with the prior single-center RCT DTR study, during which the placebo treatment did not induce a placebo effect in the control subjects [13]. The lack of change to the control group’s Disclusion Times by not undergoing real ICAGD, did not produce any control subject perceptions that they had received “a potentially corrective treatment” for their TMD symptoms (Figure 17). The placebo effect is considered real, in that it has been observed in some medical studies [40] and in a limited number of occlusal adjustment studies [41,42]. However, it apparently does not always occur, as in this study and the prior Disclusion Time Reduction RCT study [13], the control group gradually increased their symptomology over time, and became significantly more symptomatic during the 6 months after the placebo treatment was rendered.

The lack of any positive placebo effect from the control group may be reflective of the difficulty of applying an effective faux occlusal adjustment treatment, as it is not always possible to blind the operator, resulting in a single blind design. Some of the control subjects may have realized that the mock occlusal polishing was just that, or the faux occlusal treatment may have actually irritated some control subjects, resulting in a negative placebo effect. The significant increases in the control subjects’ Beck Depression Inventory-II mean scores and the Functional Restriction mean scores were likely due to the worsening of their physical conditions over time, as the placebo treatment did not in any way improve their occlusal contacts, nor improve their Disclusion Times. Disappointment with respect to the faux occlusal adjustments not helping the control subjects physically was reflected in an increase of the control group’s mean Beck Depression score. 

### 4.2. ICAGD Is Defined by Numerical Occlusal Adjustment Endpoints

The subjects’ reported symptom improvements were obtained by implementing two synchronized occlusal technologies that employed T-Scan HD Novus sensors, and BioEMG III conductive adhesive bipolar electrodes, who’s datasets clinically guided five calibrated doctors to perform the same occlusal adjustment procedure on separate subject groups in different cities. The five treating practitioners ultimately obtained consistent clinical outcomes that were verified statistically, because the providers performed ICAGD to the same range of numerical short Disclusion Times and occlusal balance tolerances (Disclusion Time ≤ 0.5 s per excursion; Occlusion Time ≤ 0.2 s per intercuspated closure; 50% +/− 4% Right-to-Left occlusal force balance in MIP) [4,5,6,7,8,9,10,15,16,34,43]. As such, this RCT study mirrored the results of other multi-center non-RCT studies that statistically improved emotional well–being [12], reduced chronic cold tooth sensitivity [9], and improved Temporomandibular Joint health by two Piper Classifications [43]. 

Disclusion Time Reduction (DTR) [4] of all molars and premolars to <0.5 s per excursion has been shown repeatedly to reduce muscle hyperactivity levels and its related symptoms pre- to post-treatment, with authors reporting statistically significant differences in Disclusion Time durations, muscle contraction levels, and lessened Time-to-Muscle Shutdown [3,4,5,6,7,8,9,10,11,12,13,14,15,16,17,18]. However, when occlusal adjustment studies are performed without the T-Scan technology using solely articulating paper markings, the occlusal adjustments are unmeasured and subjective, being absent of any occlusal contact force and timing measurements. Studies show articulating paper marks offer no information regarding which contacts bear heavier or lighter forces, nor can paper markings describe contact timing [44,45,46]. Further, three separate studies indicate dentists subjectively interpret paper markings very poorly, only succeeding in selecting correct forceful contacts 4.5–13.3% of the time, while choosing incorrect contacts 86.7–95.5% of the time [47,48,49]. This subjective paper mark/contact force interpretation procedure was likely a factor in why authors of non-T-Scan-based TMD occlusal adjustment studies failed to resolve occlusal dysfunction symptoms [27,28,29,42].

### 4.3. T-Scan High Definition Novus Sensor Demonstrate Repeatability and Durability

T-Scan HD Novus sensors (Figure 1 and Figure 2) record unique occlusal contact force and timing data from within contacting and excursing opposing teeth, which cannot be obtained in any other way. The HD sensor fits easily between opposing dental arches, whereby occlusal contacts present when a patient closes into Maximum Intercuspation (MIP) or excurses in lateral mandibular movements, and captures frictional contacts in real-time, that activate their sensels under loading. The T-Scan companion software makes it possible for a clinician to accurately diagnose and treat both the occlusal force component and the timing of the loading that is present in a patient’s malocclusion. No other occlusal indicator has been shown capable of repeatedly capturing this specific relative occlusal force and timing data from within the dental arches [50], while maintaining sensor structural integrity despite being repeatedly crushed between teeth. The 100 um Novus HD sensor thickness is a positive sensor attribute, as it houses and protects sophisticated printed electronic components within a flexible and compressible Mylar substrate. The HD sensor electronically measures 256 occlusal force levels and contact timing data accurately and repeatedly, and it has been shown in both research and clinical papers to continuously report force and timing data when used repeatedly, without sustaining significant sensor damage breakdown from intercuspating teeth [3,4,5,6,7,8,9,10,11,12,13,14,15,16,17,18,50,51,52,53]. A 2006 force reproduction study clearly showed the 100-um sensor thickness was not a factor that influenced the HD sensor’s ability to repeatedly and consistently report multiple relative occlusal force levels using multiple differing HD sensors [51]. To date, no T-Scan author has reported “frequent sensor perforation” caused by occlusal damage as being a problem when recording with Novus HD sensors.

A recent Systematic Review stated that much scientific evidence supported using the T-Scan technology in dental treatment because it measures relative occlusal contact forces and timing objectively, accurately, and repeatedly [50]. Validity studies of the T-Scan HD sensor were performed in 2006 [51], 2010 [52], 2012 [53], and 2014 [54], all of which indicated that the HD sensor accurately measured relative occlusal contact force levels in multiple locations from within contacting dental arches.

### 4.4. The EMG Electrodes Work in Concert with T-Scan Sensors to Measure the Occluso-Muscle Neurophysiology

The EMG electrodes and the BioPak software recorded the muscle physiology consequences that directly result from the quality of all subjects’ occlusal function. The EMG sensors captured the muscle physiology improvements and lower contractions levels the ICAGD coronoplasty effected in the bilateral temporalis and masseter muscles of the treatment group, while also capturing and documenting the lack of muscle physiology changes within the control subjects that underwent no occlusal changes. It can be argued that the synchronized use of T-Scan HD sensors for the occlusal function, combined with the BioPak EMG electrodes, represent the state-of-the-art in occlusal function diagnosis and treatment. 

### 4.5. Limitations

A possible limitation was that the five calibrated ICAGD clinicians made all the pre-ICAGD and post-ICAGD recordings and performed all of the ICAGD treatments on all of the subjects. Different clinicians would be expected to produce somewhat different ICAGD endpoints, (which they did obtain based upon the varying reported symptom changes) rather than obtaining exactly the same end-results, subject to subject. As the groups were also different, expecting the same outcome is not warranted. ICAGD treatment end-result precision depends on (a) the skill of the clinician performing ICAGD, (b) his/her skill level working with T-Scan excursive movement Disclusion Time datasets, and (c) on proper patient selection, as no two groups of 10 TMD patients are ever identical in all occlusal aspects. Varying ICAGD outcome precision could possibly affect the degree of subject response to ICAGD in a different clinician’s subject pool. However, the numerical datasets obtained from each separate treating clinician post-ICAGD were quite similar, indicating that calibrating the clinicians minimized the clinician variability and ensured similar occlusal adjustment outcomes. Therefore, this study’s results may not be representative of all outcomes by all clinicians, who may practice without undergoing definitive, proper TScan/EMG and ICAGD training and ICAGD coursework.

A second possible limitation was that the treatment group was older and significantly more symptomatic than the control group, which was not the problem that it could have been had the subject groups been reversed.

## 5. Conclusions

This multi-center study indicated that multiple different doctors using multiple different Novus HD T-Scan sensors could measurably perform ICAGD treatment and meet ICAGD’s known numerical tolerances, which reduced many chronic painful muscular TMD symptom intensities and frequencies, functional restrictions, and levels of emotional depression. However, in none of the five treatment centers did placebo polishing initiate a placebo effect in the control subjects, because no physical changes to the Disclusion Times within the control group were made. Importantly, the physiologic, numerical outcome measures observed were only made possible because the occlusal sensors and electromyography electrodes accurately recorded and reported statistically significant changes in the treatment group’s occlusal function and the muscle contraction levels, while recording and reporting no significant changes in the control subjects.

## Figures and Tables

**Figure 1 sensors-21-07804-f001:**
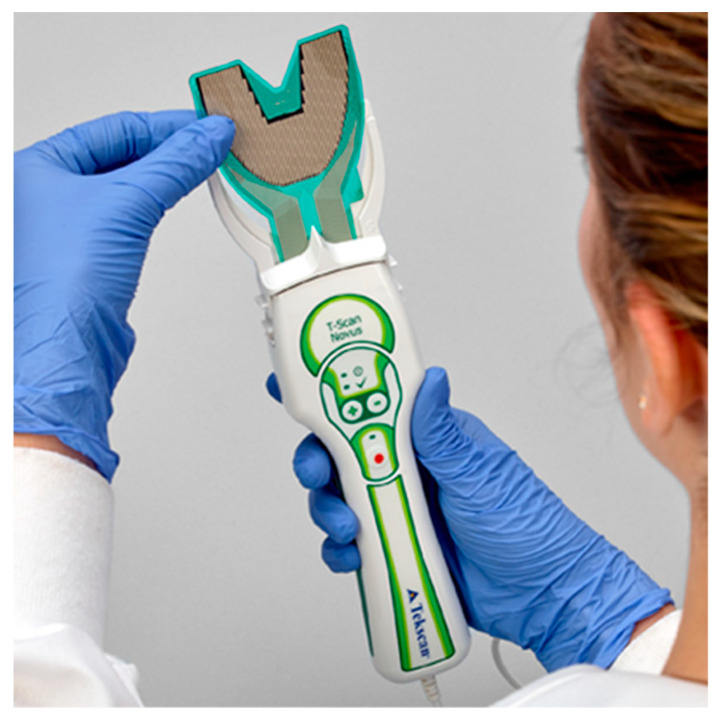
The T-Scan HD Novus sensor being inserted into the Novus recording handpiece. The HD sensor is an ultra-thin, flexible printed circuit that records 256 relative occlusal force levels and occlusal contact timing sequences when crushed between teeth.

**Figure 2 sensors-21-07804-f002:**
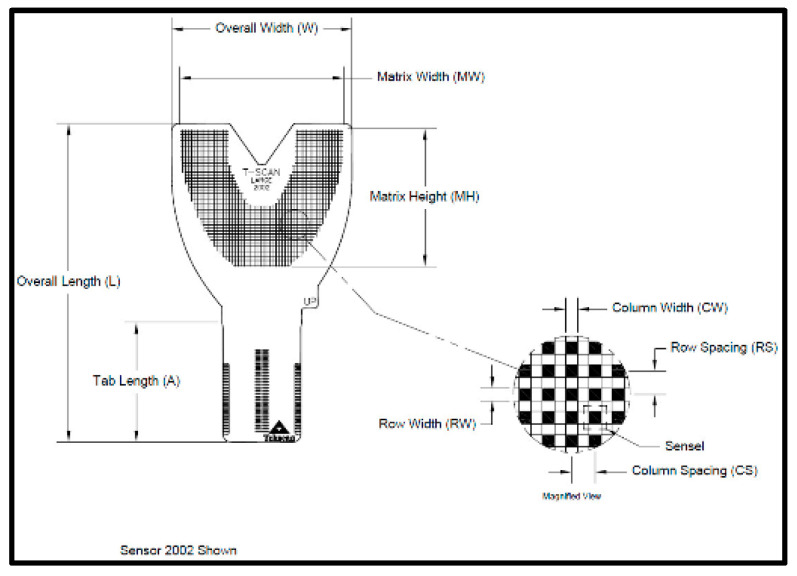
The HD sensor structural schematic illustrating the 1.61 mm^2^ sensel row and column conductive ink grid shaped in a dental arch.

**Figure 3 sensors-21-07804-f003:**
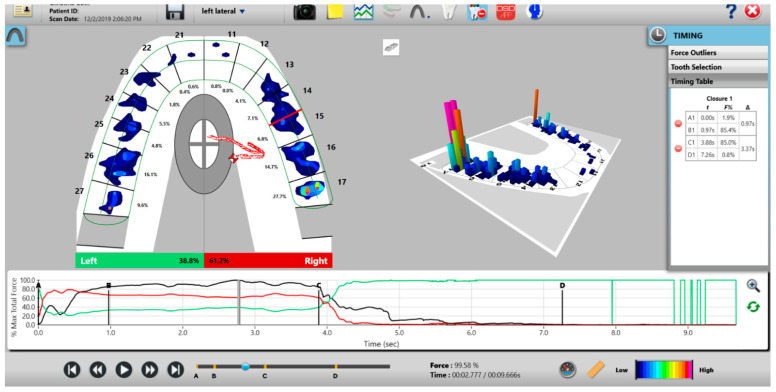
The T-Scan desktop has both a two- and three-dimensional display that reports 256 levels of occlusal force in a color-coded dynamic video format, which dentists can employ to diagnose and treat differing occlusal force and timing abnormalities.

**Figure 4 sensors-21-07804-f004:**
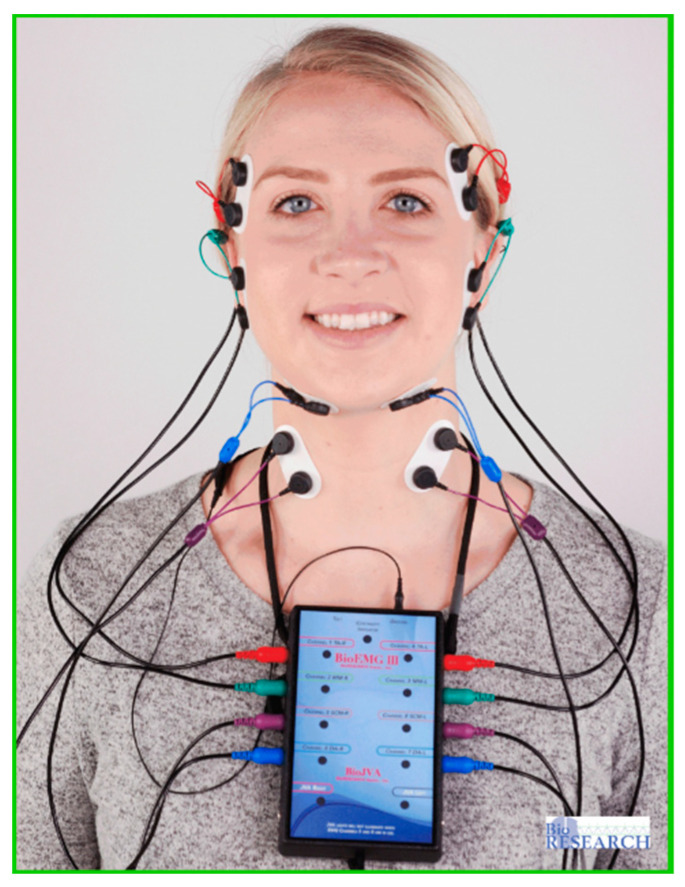
The BioEMG III system with eight muscle leads connected to simultaneously record the bilateral temporalis (red leads), masseter (green leads), sternocleidomastoid (purple leads), and digastric (blue leads) muscles.

**Figure 5 sensors-21-07804-f005:**
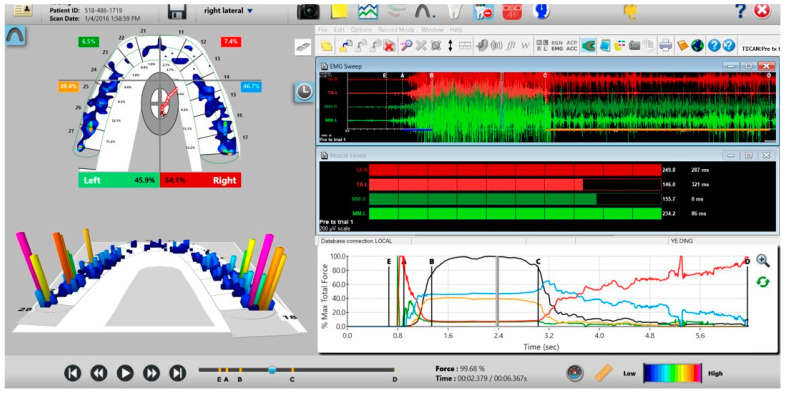
The T-Scan 10/BioEMG III desktop simultaneously displays the interrelationship between the occlusal function (T-Scan—left pane; Force vs. Time Graph—bottom right pane) and four masticatory muscles’ contraction physiology caused by the occlusal function (BioEMG III; top and middle right panes).

**Figure 6 sensors-21-07804-f006:**
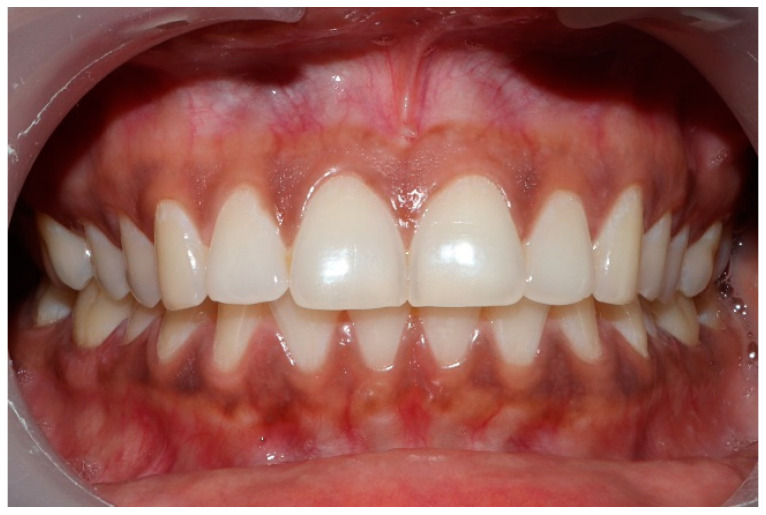
A treatment subject (who’s case will be followed throughout this manuscript) in Maximum Intercuspation (MIP). Note the worn maxillary canines, which predisposed the subject to shallow lateral anterior guidance angles that resulted in posterior prolonged Disclusion Times and high excursive muscle activity levels.

**Figure 7 sensors-21-07804-f007:**
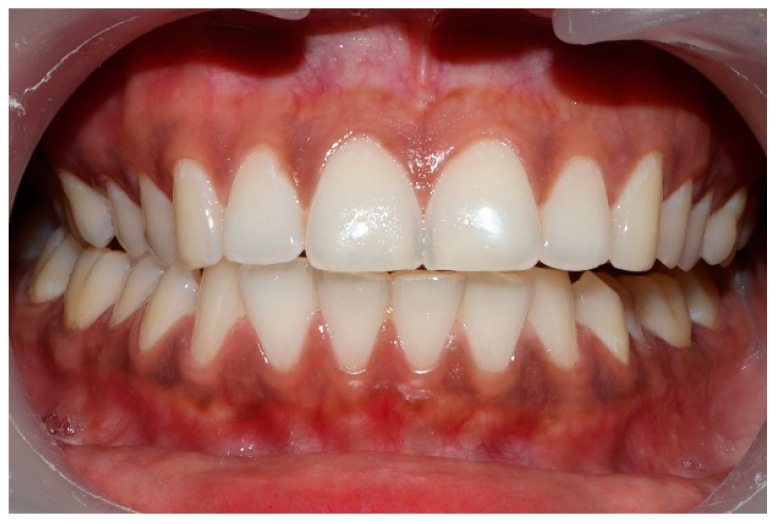
The right lateral movement during which the posterior right teeth engage because the worn right canine does not lift the right posterior teeth apart.

**Figure 8 sensors-21-07804-f008:**
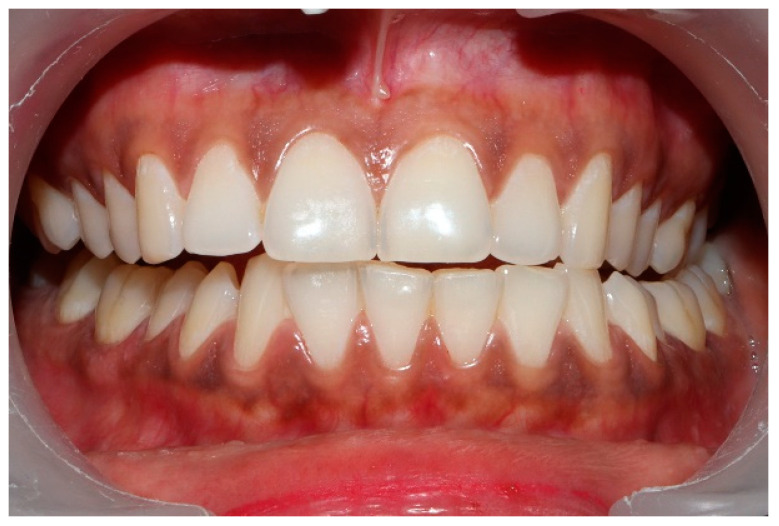
The left lateral movement with visible working side group function contacts between the mandibular 2nd and 3rd molars and the maxillary 1st and 2nd molars. The worn maxillary left canine #11 offers shallow lift that does not separate the opposing posterior left teeth.

**Figure 9 sensors-21-07804-f009:**
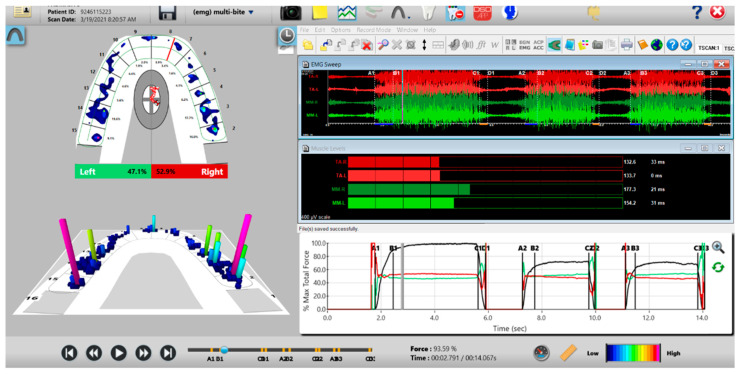
A multi-bite recording showing near equal right-to-left side occlusal balance in MIP (47.1% left, 52.9% right). Although both temporalis and masseter muscles contract similarly, in the 2nd and 3rd closures the patient successively weakens (Force vs. Time graph; bottom right pane; curved black line) as the subject’s Total Force generation declines from the 1st closure to the 3rd closure.

**Figure 10 sensors-21-07804-f010:**
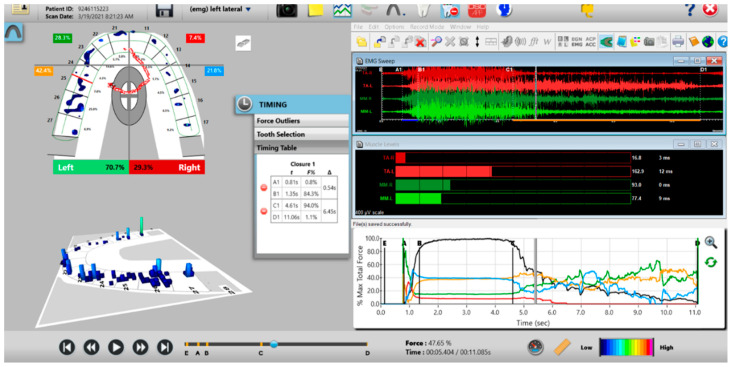
A pre-operative left excursion Disclusion Time/BioEMG recording showing (in the Force vs. Time Graph; bottom right pane), to the right of the C Line, “steps” in the Total Force line, indicating there is marked friction present between opposing posterior occlusal surfaces that hinders free mandibular movement. In the T-Scan windows (left pane), both working group function and balancing contacts contribute to the left excursive friction (Force vs. Time Graph; bottom right pane; the blue posterior right and orange posterior left quadrant lines are elevated all the way through until the vertical D Line). The upper right EMG pane illustrates significant excursive muscle hyperactivity (to the right of line C) in the left temporalis and the right masseter muscles, with the left temporalis firing all through the duration of the left excursion (up to D). The pre-treatment left Disclusion Time is considerably prolonged = 6.45 s (Timing pane).

**Figure 11 sensors-21-07804-f011:**
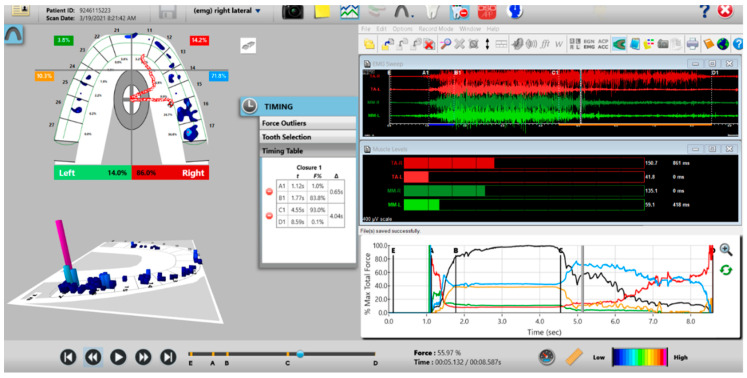
A pre-operative right excursion Disclusion Time/BioEMG recording showing (in the Force vs. Time Graph; bottom right pane), to the right of the C Line, many steps in the Total Force line, indicating there is marked right excursive friction present similar to the left excursion. In the T-Scan windows (left pane), working group function contacts control the right excursion (Force vs. Time Graph; bottom right pane; blue posterior right quadrant line rises in force and stays high all the way through until the D Line). The upper right EMG pane illustrates significant excursive muscle hyperactivity (to the right of line C) in the right temporalis and the right masseter muscles, with less in the left masseter muscle, with the right temporalis firing all through the right excursion (up to D). The pre-treatment right Disclusion Time is considerably prolonged = 4.04 s (Timing pane).

**Figure 12 sensors-21-07804-f012:**
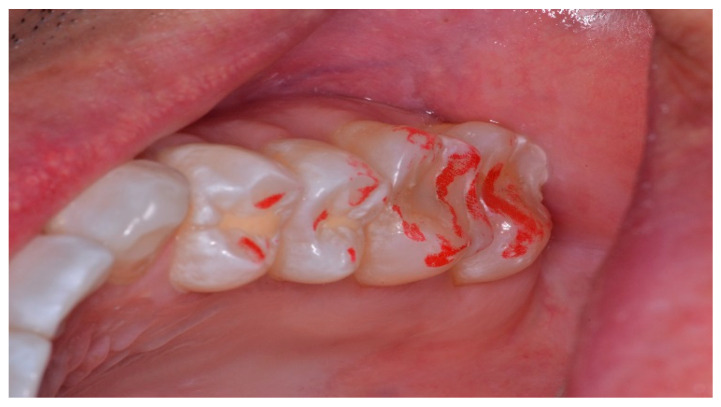
The maxillary left quadrant with many articulating paper linear markings representative of interocclusal prolonged Disclusion Time. There is wear and sharpened tooth structure visible at the distobuccal of the 2nd molar.

**Figure 15 sensors-21-07804-f015:**
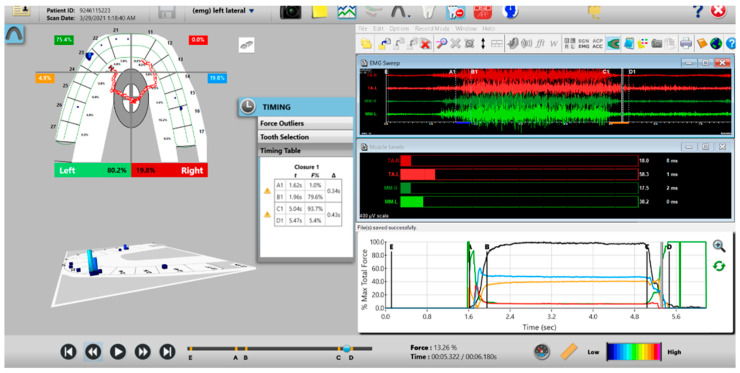
Post-ICAGD left excursive Disclusion Time/EMG recording with markedly less excursive muscle activity to the right of line C than was present pre-ICAGD (see Figure 11). The corrected function in the Force vs. Time graph (bottom right pane), no longer has friction “steps” visible after ICAGD. Short post-ICAGD Disclusion Time is denoted by the D line being very close to the C line = 0.43 s (Timing pane).

**Figure 16 sensors-21-07804-f016:**
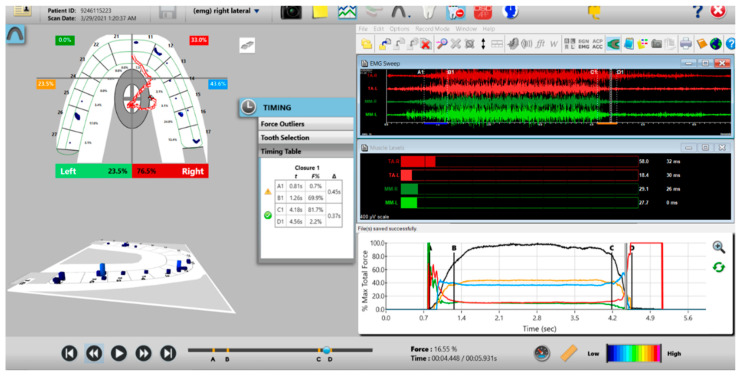
Post-ICAGD right excusive Disclusion Time/EMG recording with markedly less excursive muscle activity to the right of line C than was present pre-ICAGD (see Figure 12). The corrected function in the Force vs. Time graph (bottom right pane) no longer has friction “steps” visible after ICAGD, with short right Disclusion Time = 0.37 s (Timing pane).

**Figure 17 sensors-21-07804-f017:**
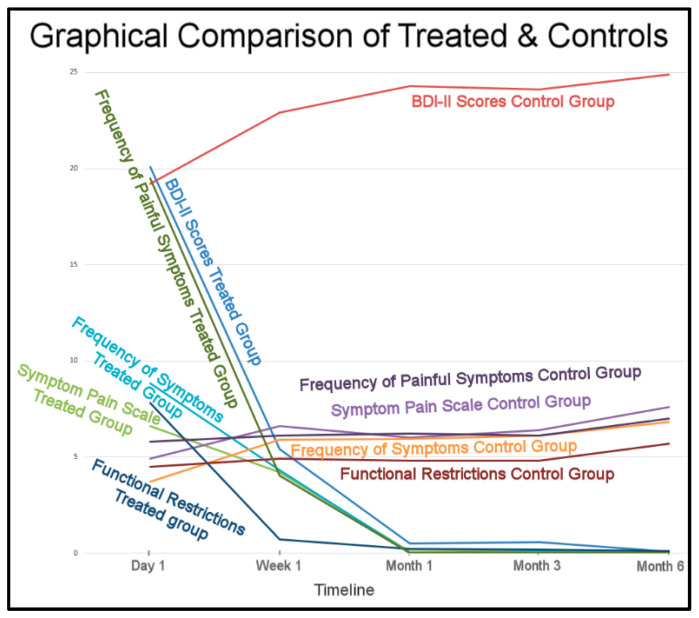
The treated subjects’ physical and emotional improvements occurred between 1 week to 1 month and remained low for 6 months. The control subjects differing symptoms maintained and worsened over the same 6 months.

**Table 1 sensors-21-07804-t001:** Pre- and post-ICAGD or Mock ICAGD Disclusion Times for all dates of measurement for both subject groups.

Comparison of Disclusion Times—Treated vs. Controls	Left Disclusion Time Pre-TX (s)	Right Disclusion Time Pre-TX (s)	Left DT Post-DTR Day 1 (s)	Right DT Post-DTR Day 1 (s)	Left DT 1 Week Post-Tx (s)	Right DT 1 Week Post-Tx (s)
	Pre-Treatment		1 Day Post-Treatment		1 Week Post-Treatment	
Treatment Group	2.99	2.75	0.33	0.33	0.32	0.31
Control Group	2.87	3.02			2.95	3.25
Student’s *t* test *p*	>0.05	>0.05			<0.05	<0.05
	Left DT 1 month Post-Tx (s)	Right DT 1 month Post-Tx (s)	Left DT 3 months Post-Tx (s)	Right DT 3 months Post-Tx (s)	Left DT 6 months Post-Tx (s)	Right DT 6 months Post-Tx (s)
	1 month Post-Treatment		3 months Post-Treatment		6 months Post-Treatment	
Treatment Group	0.30	0.30	0.30	0.29	0.28	0.27
Control Group	2.98	3.17	2.93	3.09	3.05	3.15
Student’s *t* test *p*	<0.05	<0.05	<0.05	<0.05	<0.05	<0.05

**Table 2 sensors-21-07804-t002:** Comparison of Beck Depression Inventory-II scores between the treated group and the control group at each time-point.

Beck Depression Inventory-II Scores		Treated Group Scores (0–63)	Control Group Scores (0–63)
Day 1	Mean	20.1	19.2
(Pre-treatment)	SD	4.98	7.26
	Median	21	18
Initial relation of treated to controls		*p* > 0.05	

1 week	Mean	5.4	22.9
(Post-treatment)	SD	6.93	6.56
	Median	4	23
Treated to controls at 1 week		*p* < 0.00001	
1 month	Mean	0.48	24.3
(Post-treatment)	SD	1.45	6.79
	Median	0	23
Treated to controls at 1 month		*p* < 0.00001	
3 months	Mean	0.56	24.1
(Post-treatment)	SD	0.51	7.73
	Median	0	23
Treated to controls at 3 months		*p* < 0.000013	
6 months	Mean	0.06	24.9
(Post-treatment)	SD	0.24	6.02
	Median	0	24
Treated to controls at 6 months		*p* < 0.00001	

**Table 3 sensors-21-07804-t003:** Comparison of Symptom Pain Scale scores between the treated group and the control group at each time-point.

Symptom Pain Scale Levels (0–4)		Treated Group Scores (0–48)	Control Group Scores (0–48)
Day 1	Mean	6.6	4.9
(Pre-treatment)	SD	5.96	4.28
	Median	5	5
Initial relation of treated to controls		*p* > 0.05	
1 week	Mean	4.2	6.6
(Post-treatment)	SD	2.51	5.21
	Median	4	6
Treated to controls at 1 week		*p* < 0.0056	
1 month	Mean	0.06	6.0
(Post-treatment)	SD	0.24	5.20
	Median	0	5
Treated to controls at 1 month		*p* < 0.00001	
3 months	Mean	0.16	6.4
(Post-treatment)	SD	0.51	5.24
	Median	0	6
Treated to controls at 3 months		*p* < 0.00001	
6 months	Mean	0.08	7.6
(Post-treatment)	SD	0.27	4.99
	Median	0	7
Treated to controls at 6 months		*p* < 0.00001	

**Table 4 sensors-21-07804-t004:** Comparison of Frequency of Symptoms Scale scores between the treated group and the control group at each time-point.

Frequency of Symptoms Scale (0–3)		Treated Group Scores (0–33)	Control Group Scores (0–33)
Day 1	Mean	8.84	3.7
(Pre-treatment)	SD	4.10	3.78
	Median	9	3
Initial relation of treated to controls		*p* < 0.05	
1 week	Mean	4.3	5.9
(Post-treatment)	SD	2.75	5.01
	Median	4	5
Treated to controls at 1 week		*p* = 0.1868	
1 month	Mean	0.02	5.92
(Post-treatment)	SD	0.14	4.99
	Median	0	5
Treated to controls at 1 month		*p* < 0.00001	
3 months	Mean	0.06	6.1
(Post-treatment)	SD	0.24	5.03
	Median	0	5
Treated to controls at 3 months		*p* < 0.00001	
6 months	Mean	0.04	6.8
(Post-treatment)	SD	0.198	4.89
	Median	0	6
Treated to controls at 6 months		*p* < 0.00001	

**Table 5 sensors-21-07804-t005:** Comparison of Functional Restriction scores between the treated group and the control group at each time-point.

Functional Restrictions (0–3)		Treated Group Scores (0–27)	Control Group Scores (0–27)
Day 1	Mean	7.8	4.5
(Pre-treatment)	SD	2.73	2.95
	Median	7	4
Initial relation of treated to controls		*p* < 0.05	
1 week	Mean	0.7	4.9
(Post-treatment)	SD	1.41	3.05
	Median	0	4
Treated to controls at 1 week		*p* < 0.00001	
1 month	Mean	0.22	4.8
(Post-treatment)	SD	0.74	2.99
	Median	0	4
Treated to controls at 1 month		*p* < 0.00001	
3 months	Mean	0.18	4.8
(Post-treatment)	SD	0.6	3.01
	Median	0	0
Treated to controls at 3 months		*p* < 0.00001	
6 months	Mean	0.12	5.7
(Post-treatment)	SD	0.39	2.73
	Median	0	6
Treated to controls at 6 months		*p* < 0.00001	

**Table 6 sensors-21-07804-t006:** Comparison of Frequency of Painful Symptoms scores between the treated group and the control group at each time-point.

Frequency of Painful Symptoms (0–3)		Treated Group Scores (0–30)	Control Group Scores (0–30)
Day 1	Mean	19.5	5.8
(Pre-treatment)	SD	5.51	4.65
	Median	20	5
Initial relation of treated to controls		*p* < 0.05	
1 week	Mean	4.04	6.1
(Post-treatment)	SD	1.67	4.68
	Median	4	5
Treated to controls at 1 week		*p* = 0.0043	
1 month	Mean	0.02	6.2
(Post-treatment)	SD	0.14	4.57
	Median	0	5
Treated to controls at 1 month		*p* < 0.00001	
3 months	Mean	0.02	6.1
(Post-treatment)	SD	0.14	4.66
	Median	0	5
Treated to controls at 3 months		*p* < 0.00001	
6 months	Mean	0.02	6.98
(Post-treatment)	SD	0.14	4.25
	Median	0	7
Treated to controls at 6 months		*p* < 0.00001	

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
