# Peer review of "A Multi-Center Disclusion Time Reduction (DTR) Randomized Controlled Occlusal Adjustment Study Using Occlusal Force and Timing Sensors Synchronized with Muscle Physiology Sensors"

_sensors, 2021, doi:10.3390/s21237804_

Round 1
Reviewer 1 Report
Please think of a title that include sensors in it or mention the technology used in research
Fig 17 seems to belong to results
There is no mention of EMG recordings at discussion
Please include a conclusion on sensors
Please ad to reference list some papers on t-scan/EMG published in the last 5 years by other authors
Author Response
- Please think of a title that include sensors in it or mention the technology used in research -
A Multi – Center Disclusion Time Reduction (DTR) Randomized Controlled Occlusal Adjustment Study Using Occlusal Force and Timing Sensors Synchronized with Muscle Physiology Sensors
2. Fig 17 seems to belong to results - Figure 17 moved to the Results, and referred to in the Discussion with regards to the lack of placebo effect in the control subjects
3. There is no mention of EMG recordings at discussion - Below was added to the Discussion
4.4 The EMG Electrodes work in concert with T-Scan Sensors to measure the Occluso-muscle neurophysiology
The EMG electrodes and the BioPak software recorded the muscle physiology consequences that directly result from the quality of all subjects’ occlusal function. The EMG sensors captured the muscle physiology improvements and lower contractions levels the ICAGD coronoplasty effected in the bilateral temporalis and masseter muscles of the treatment group, while also capturing and documenting the lack of muscle physiology changes within the control subjects that underwent no occlusal changes. It can be argued that the synchronized use of T-Scan HD sensors for the occlusal function combined with the BioPak EMG electrodes, represent the State of the Art in occlusal function diagnosis and treatment.
4. Please include a conclusion on sensors - Added to the Conclusions
Importantly, the physiologic, numerical outcome measures observed were only made possible because the occlusal sensors and electromyography electrodes accurately recorded and reported statistically significant changes in the treatment group’s occlusal function and the muscle contraction levels, while reporting no significant changes in the control subjects.
5. Please add to reference list some papers on t-scan/EMG published in the last 5 years by other authors
Sierpinska T, Kuc J, Golebiewska M. Assessment of masticatory muscle activity and occlusion time in patients with advanced tooth wear. Archives of Oral Biology, 2015;60:1346-1355
Stocka A, Sierpinska T, Kuc J, Golebiewska M. Relationship between depression and masticatory muscles function in a group of adolescents. The Journal of Craniomandibular & Sleep Practice, 2018;36:390-399
Sierpinska T. Tooth Wear – Prevention, Treatment and Monitoring using the T-Scan/BioEMG Synchronization Module. In Handbook of Research on Clinical Applications of Computerized Occlusal Analysis in Dental Medicine. 2020. Pp.879-944. Hershey PA, IGI Global Publishers. http://doi:10.4018/978-1-5225-9254-9
Reviewer 2 Report
The manuscript is very well written and deals with a relevant topic of great importance to the adult population. Some doubts persisted and need to be clarified in the text - one of the potential biases is the use of other complementary therapies to control symptoms - pain, functionality. For example - use of medications and other alternative therapies - was these variables controlled? Have any other potentially uncontrolled interaction or confusion variables been identified?
As the analysis was carried out in the follow-up (time), was the relative adjustment made for it? If not, why?
Figure 17 is unclear and expendable.
Author Response
1. No complimentary therapies and no pain medications were applied to either the control group or the treatment group. The only treatments applied were the ICAGD coronoplasty to the treated group, and tooth polishing as the "placebo ICAGD" to the Control Group. No other uncontrolled variables were identified.
The above text was added to the Method and Materials at the end of section 2.6
2. Even though patients received treatment on different days, the analyses were done first at one week to assess the immediate response to treatment, and then the analyses were repeated at 1 month, 3 months and 6 months from the starting date of each patient to assess whether the response was sustained.
Since all of the patients in the treatment group, and likewise in the control group, were assessed at the same relative timepoints, no time adjustment was indicated.
The above text was added to the Method and Materials section 2.7
Reviewer 3 Report
First of all, please shorten also the abstract. It stays written in instruction for authors that:” The abstract should be a total of about 200 words maximum”. You should also add a “background” section. My advice is to achieve this by reducing the volume of materials and methods and a little bit of results section (e.g. delete p<0,05).
Please shorten also the introduction. This part of the article should explain the basic concepts, i.e. the initial description of the equipment, the background of the research problem, a brief description of the research problem and the purpose of the work. Many of its paragraphs [96-101, 118-128, 143-147, 168-176] should be a part of the discussion, because they deal with factors that may affect the outcome of the research. Their direct possible impact on the results should also be also further discussed.
Do you have any explanation why most of the symptoms in the study group disappeared after the first month? If so, please provide a more detailed explanation of the timing in the discussion. I have the impression that this topic has not been adequately covered.
You can also include in the discussion the topic of the use of AI in diagnostics. You can find useful information in the articles:
Bernauer SA, Zitzmann NU, Joda T. The Use and Performance of Artificial Intelligence in Prosthodontics: A Systematic Review. Sensors (Basel). 2021;21(19):6628. OR Askar H, Krois J, Rohrer C, Mertens S, Elhennawy K, Ottolenghi L, Mazur M, Paris S, Schwendicke F. Detecting white spot lesions on dental photography using deep learning: A pilot study. J Dent. 2021 Apr;107:103615.
The reference style does not follow the MDPI/ACS style recommended by the journal. Please change it. The reference in text should be in square brackets.
You uploaded the file with the co-authors comments. Why?
Please, before resubmission, read the instructions for authors carefully. The article is interesting and one can see the work put into this study. However, in some places, the manuscript looks like a sloppy transfer, without proper respect towards the journal.
I discourage the acceptance of the manuscript in the present form.
Author Response
1. The Abstract was reduced to 204 words with the headings Objective, Method and Materials, Results, Conclusions
Without the headings, the Abstract is 199 words
2. Please shorten also the introduction. This part of the article should explain the basic concepts, i.e. the initial description of the equipment, the background of the research problem, a brief description of the research problem and the purpose of the work.
The original Introduction is now preceded by a Background section (1.1), that describes the sensors used in this research, and how they record and numerically report human occlusal function and muscle physiology. The Introduction (1.2) follows the Background where the research problem and the purpose of the study are outlined. The Introduction was shortened by moving paragraphs defined by lines 96-101, 118-128, 143-147, 168-176 to the Discussion, with edits made for continuity and brevity. The citations were renumbered to reflect moving this content. Additionally, the authors have attempted throughout the Introduction and Background to compress the text and remove any overly wordy paragraphs.
The authors however, respectfully decline to overly shorten the Method and Materials and the Results. Successful future replication of this research warrants the detailed descriptions within the Method and Materials, that accurately report how the clinical procedures were performed, and detail how to analyze the key parts of the T-Scan 10/BioEMG III data sets. This will make it possible for other ICAGD-trained DTR researchers to replicate this study’s methodology, so that future Multi-center or single center ICAGD/DTR treatment study outcomes could more adequately be compared to this Multi-center ICAGD/DTR research. As for shortening the Results, presently Tables 1-6 and Figure 7 concisely document the entirety of the physiologic changes the treated subjects experienced, and the control subjects did not experience. Again, the authors have attempted throughout the Method and Materials and the Results to compress the text and remove any overly wordy paragraphs.
3. With respect to "brackets" for citations, the authors were sent a Sensors Journal Template to use for the original submission, which contained parentheses for citations, that the authors utilized. The resubmission document sent to the authors for Round 1 revisions (sensors-1445736) has continued the use of parentheses and not brackets.
An example of a citation in the revision document from page 3 uses parentheses: "Higher occlusal forces give off higher measured digital output voltages, while lower the applied forces give off lower digital output voltages.(1)
So unless the authors are notified by the Sensors Journal office or the Head Editor of this Special Issue, the authors will use the citation format in the revisions document sensors-1445736.
4. A detailed explanation of the rapid symptom resolution following ICAGD has been added to the Discussion.
Symptom resolution for most treated subjects occurred within the 1st month after their long Disclusion Times were reduced to short Disclusion Times. Symptoms rapidly lessened because the pre-treatment elevated levels of muscle activity that cause lactic acid buildup, and its’ resultant ischemic pain and functional limitations of the masticatory muscles, was markedly lessened after ICAGD removed the occlusal surface friction from the subject’s mandibular excursions. These physiologic changes are neurologically-mediated within the Central Nervous System(34), because the time-duration and volume of posterior teeth pulpal nerve fiber flexures and PDL nerve fiber compressions, are both drastically reduced by ICAGD, making the treated subjects function muscularly at or near resting state contraction levels.(4-6). These low-level excursive movement contractions stop lactic acid from building up, which limits the resultant ischemia, allowing for the lactic acid to be metabolized away, instead of it continuing to accumulate. Without ischemia present, the masseter and temporalis muscle fibers can re-oxygenate, which lessens muscular pain, reduces the frequency of symptom appearances, and resolves functional limitations, whereby chewing function dramatically improves.(3) This ICAGD-induced absence of muscular ischemia is quickly followed by lasting symptom resolution.(10,17,40).
5. The reviewer’s suggestion to discuss the use of “AI diagnostics” appears to contradict his/her request to shorten the manuscript. Especially, in that this study had no AI technology used in it, and to date, no AI application has been developed for diagnosing Long Disclusion Time, occlusal surface excursive friction, and masticatory muscle hyperactivity. The ICAGD coronoplasty is a manually performed enamel removal procedure, which could possibly in the future, be guided by AI navigational technology (like a surgical procedure can be navigationally-guided), but at present that’s not possible. Instead, this study documented how 5 treating practitioners were guided by the T-Scan pre-operative Long Disclusion Time data to adjust the treated group’s long Disclusion Times down to numerically short Disclusion Times with high numerical and physiologic tolerances that resolved many symptoms.
The authors respectfully decline to discuss “AI diagnostics” to minimize manuscript length because no current-day AI diagnostic application exists for Disclusion Time Reduction (DTR).
Round 2
Reviewer 3 Report
Dear authors,
I had no intention of abbreviating your manuscript at all. A request for changes results from the instructions available at the link: https://www.mdpi.com/journal/sensors/instructions , which, given the wrong citation format change, seems strange to you. I am very pleased with the amendments introduced by you and introduciton of the explanation, that I asked for. The article is now clearer and much more easier to read.
It is a pity that you did not include the topic of AI in the discussion, although I hope you will include it in future research. The article itself is of a high standard and I am glad to participate in its review process.
I recommend the acceptence of the manuscript after changing the citation format.
Author Response
- All citation numbers are now bracketed inside the punctuation [52],
An example of a bracketed citation in the revision document (Sensors -1445736 (2)) from page 3:
"Higher occlusal forces give off higher measured digital output voltages, while lower the applied forces give off lower digital output voltages [1].”
- And it’s quite possible in future DTR research studies, that the implementation of “AI diagnostics” will be explored.